# A Compact 16 Channel Embedded System with High Dynamic Range Readout and Heater Management for Semiconducting Metal Oxide Gas Sensors

**Christof Hammer [1],\*, Johannes Warmer [2] , Stephan Maurer [2], Peter Kaul [2] , Ronald Thoelen [3] and Norbert Jung [2]**

[1]    Institute for the Protection of Terrestrial Infrastructures, German Aerospace Center, Rathaus Allee 12, 53757 Sankt Augustin, Germany

[2]    Institute of Safety and Security Research ISF, University of Applied Sciences Bonn-Rhine-Sieg, Grantham Allee 20, 53757 Sankt Augustin, Germany; johannes.warmer@h-brs.de (J.W.); stephan.maurer@h-brs.de (S.M.); peter.kaul@h-brs.de (P.K.); norbert.jung@h-brs.de (N.J.)

[3]    Institute for Materials Research, Hasselt University, Wetenschapspark 1, B-3590 Diepenbeek, Belgium; ronald.thoelen@uhasselt.be

\*    Correspondence: christof.hammer@dlr.de; Tel.: +49-2241-20148-31

**Abstract:** The simultaneous operation of multiple different semiconducting metal oxide (MOX) gas sensors is demanding for the readout circuitry. The challenge results from the strongly varying signal intensities of the various sensor types to the target gas. While some sensors change their resistance only slightly, other types can react with a resistive change over a range of several decades. Therefore, a suitable readout circuit has to be able to capture all these resistive variations, requiring it to have a very large dynamic range. This work presents a compact embedded system that provides a full, high range input interface (readout and heater management) for MOX sensor operation. The system is modular and consists of a central mainboard that holds up to eight sensor-modules, each capable of supporting up to two MOX sensors, therefore supporting a total maximum of 16 different sensors. Its wide input range is archived using the resistance-to-time measurement method. The system is solely built with commercial off-the-shelf components and tested over a range spanning from 100 Ω to 5 GΩ (9.7 decades) with an average measurement error of 0.27% and a maximum error of 2.11%. The heater management uses a well-tested power-circuit and supports multiple modes of operation, hence enabling the system to be used in highly automated measurement applications. The experimental part of this work presents the results of an exemplary screening of 16 sensors, which was performed to evaluate the system's performance.

**Keywords:** high dynamic range resistance readout; semiconducting metal oxide gas sensor array; automated sensor-screening; high diagnostic coverage and reliability

## 1. Introduction

Metal Oxide (MOX) gas sensors are cheap, easy to manufacture, and have therefore become an economic success in many applications such as gas leak detection or air quality monitoring [1]. Research projects have shown that commercially available sensors can also easily be used for the detection of substances other than those they were originally designed to detect [2].

In such scenarios, however, the sensors resistive output can, depending on the target gas and sensor material, easily have a dynamic range of several decades. To determine which types of substrates can be used for which types of gas at what sensor temperature, they have to be experimentally characterized in detail. This need for characterization especially applies to newly

developed custom-made sensor coatings in order to evaluate their performance for both new as well as known applications [3].

The mentioned characterization is known as sensor-screening and can be a very time-consuming task [4,5]. Therefore, a high degree of automation for this process is very desirable. One method for speeding up the screening process is parallelization: Multiple (different) sensors are exposed to the same target simultaneously. Their resistive values are captured, and, after a predetermined time, their substrate temperature is changed. This procedure is repeated for all programmed temperature steps.

Having different types of sensors that are all exposed to the same target at the same time is very challenging for the readout equipment. This challenge results from some sensors having only small resistive variations at a low resistive baseline (e.g., several 100 Ω) while others change their resistance over a span of six decades or more. Therefore, an adequate readout system must provide a sufficient dynamic range, to support all kinds of sensors (and sensor combinations) attached to it.

In the following, we present a compact and modular embedded system that can simultaneously read and supply up to 16 MOX gas sensors. The system covers a wide dynamic measurement range of 9.4 decades spanning from 10 Ω to 4 GΩ. Each sensor's heater element is powered by a dedicated digitally programmable power supply, which supports several operating modes such as constant temperature or temperature cyclic operation.

## 2. Related Work

Considering that this work's main contribution is situated in the field of complete interfaces for MOX gas sensors and MOX sensor arrays comprising sensor readout and heater management, we investigated the approaches that have been used and proposed by other research teams.

In 2005, Grassi et al. built a high-precision wide-range front-end for resistive gas sensors arrays [6]. This system is based on a transimpedance amplifier with programmable amplification, fixed excitation voltage, and a 13-Bit Analog to Digital Converter (ADC). Its dynamic range is given as 100 Ω to 10 MΩ. Later, in 2006 and 2007, Grassi presented two new measurement circuits, now implementing the resistance-to-time measurement principle, spanning an input range from 1 kΩ to 1 GΩ. [7,8]

Then, in 2008, Lombardi et al. presented a fully integrated interface for MOX sensors [9]. Similar to Grassi's work, the system uses the resistance-to-time principle. While the dynamic input range is slightly smaller, spanning only 5.3 decades (10 kΩ to 2 GΩ), it is enhanced by an integrated temperature management circuit, providing a single chip solution for the management of a single MOX sensor.

In the following years, both the measurement principle and the heater management techniques have been enhanced and were integrated into the various designs [10–15].

In December 2019, Ciciotti et al. presented "A 450 μA 128-dB Dynamic Range A/D CMOS Interface for MOX Gas Sensors" [16]. Similar to prior works, they built the circuit with a range from 100 Ω to 1 MΩ. They claimed to have a large linearity without glitches over a resistance range from 100 Ω to 1 MΩ with a maximum relative error of 0.4%.

All researched works have the common idea to integrate the wide range readout for a single MOX gas sensor into a custom mixed-signal application-specific integrated circuit (ASIC). In a few of the designs, the heater management is also addressed but only in a very basic way (e.g., by using duty cycle regulation of a square wave). While all of the works produced specific CMOS chips, these are mostly prototypes and are therefore not commercially available, effectively limiting their practical use to the respective research group.

This work presents a modular embedded system that can be constructed with commercial off-the-shelf (COTS) components. It implements a wide dynamic range readout over 9.4 decades using the resistance-to-time measurement principle. Each sensor module of the system fully supports up to two MOX sensors, providing independent heater control and resistive readout. Furthermore, programmable sensor excitation voltages are supported for each sensor. The modular design and

simple digital interface enable the system to be easily extended to support multiple sensors in an array formation.

## 3. System Design

The proposed system is designed to support in total up to 16 user-selectable MOX sensors. To avoid parasitic capacitance, leakage current, and achieve low noise operation, it was decided to place the sensors as close as possible to the input of the measurement circuit. This key requirement leads to the design decision to create a modular ring-shaped embedded system, designed around a water-cooled measurement chamber which provides the space for up to 16 sensors. Since the measurement circuitry is slightly influenced by the surrounding temperature, it was decided to equip the chamber with an active water cooling system. It prevents the heat from the sensors from reaching the printed circuit board of the measurement circuit. The system consists of a mainboard that manages power distribution, power supervision, and communication with up to eight sensor modules, which provide the interface for a maximum of two MOX sensors.

As shown in Figure 1, the mainbord's shape is a circle with an outer diameter of 96 mm and an inner diameter of 64 mm wrapping around a custom-designed, eight-sided measurement chamber. The eight sockets for the sensor modules are placed along a circular path with an angle of 45°. A mirrored copy of the mainboard's shape, created with a 3D printer, is used to affix the sensor modules on the backside of the system.

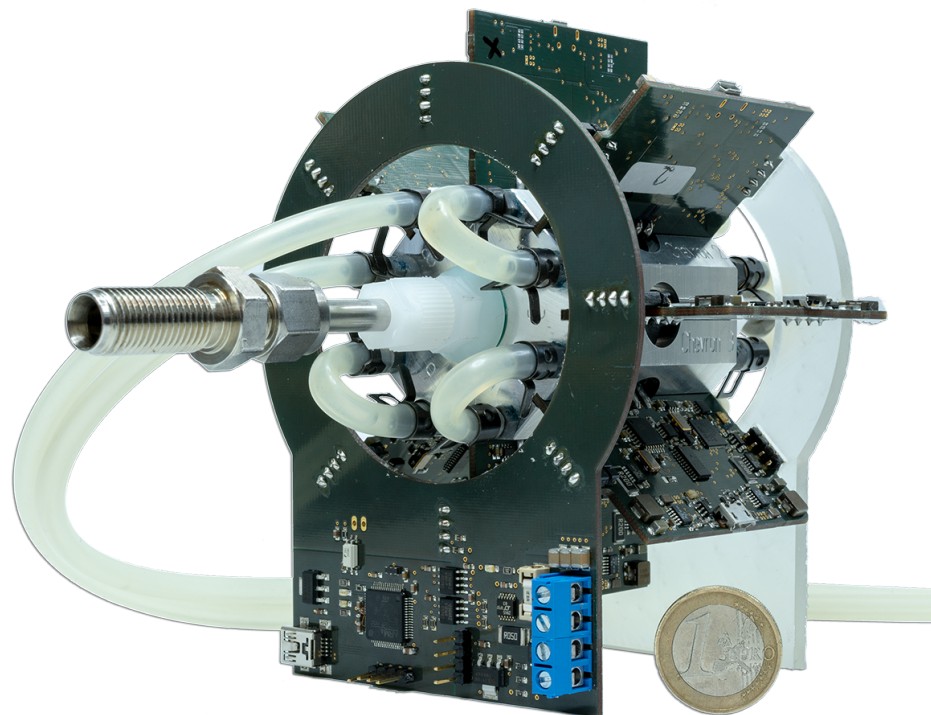

**Figure 1.** The proposed system attached to a custom-built MOX-Chamber with water cooling.

The core circuitry follows the same design and reliability principles as shown in our previous work [17] and is hence not described in detail. The central power line is protected against over-voltage, under-voltage, and reverse-voltage conditions by the the commercially available supply protection chip LTC4365 [18]. Additional continuous supervision of the system's main voltage and current ($V_{SYS}$ and $I_{SYS}$) is realized by the LTC2945 [19] power monitor, which provides a user-programmable shutdown option on top of the hardwired absolute maximum supervision performed by the LTC4365.

The main $\mu$-controller STM32F410 [20] connects the internal system communication (1 MHz $I^2C$ bus), with the external USB (Virtual Com Port) communication. The controller is electronically isolated

from the remaining mainboard components and sensor modules, protecting the attached computer (and vice versa the system) in the case of a critical electrical failure. Furthermore, the isolation provides a very effective protection against noise introduced by the USB bus. This isolation is implemented with the ADUM1250 chip [21].

To keep the interface connector for the sensor modules as small as possible, only the $I^2C$ lines and the main power line $V_{SYS}$ and system ground $GND_{SYS}$ are used. There are no additional addressing lines included, requiring each sensor board to have a unique $I^2C$ addresses coded in the firmware.

A sensor module implements the readout and heater circuit for two MOX sensors. A built prototype with a simplified block diagram that depicts the essential components is shown in Figure 2. The heater circuit is based on a well-tested design from a previous work [17] built with the LTC3600 [22] buck down converter in conjunction with the aforementioned LTC2945 and is therefore not further elaborated here. Communication and data transfer to the master, measurement operation, and temperature management is coordinated by a central STM32L433CC [23] microcontroller. The wide-range resistive readout is realized with the DDC112 [24] current-to-digital chip from Texas Instruments.

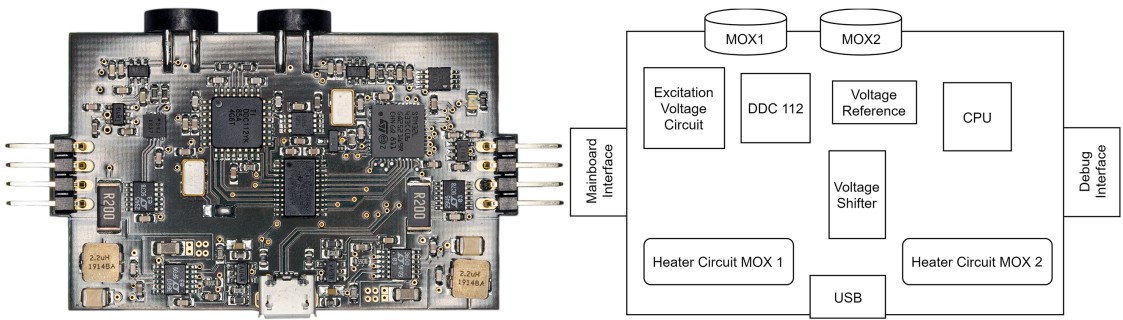

**Figure 2.** (**Left**) The built prototypical sensor module. (**Right**) Simplified floorplanning diagram showing the positions of the most important building blocks.

The DDC IC series [24] from Texas Instruments is a highly integrated current-to-digital converter, originally designed for computer tomography applications. The chip offers, depending on the model, multiple channels with selectable integration ranges connected to an internal high precision 20-Bit analog-to-digital converter (ADC). The chosen DDC112 provides two input channels, which are sampled simultaneously. Current is integrated for a user-selectable integration time $T_{Int}$ and the digitized 40-Bit data word is provided by an Serial Peripheral Interface (SPI) compatible interface.

To deliver continuous measurements, each channel consists of two identical integrators: Side A and Side B. While Side A accumulates the current, the value of Side B is digitized and vice versa. The chip has seven internal integration ranges from 50 to 350 pC and can be extended by an additional range, for up to approximately 1000 pC of charge, which is achieved with four external capacitors.

The DDCs nominal analog supply voltage is 5 V, and, since it is not equipped with a dedicated digital power interface, a voltage translator for the conversion between the DDC signals and the microcontroller's 3.3 V power supply is needed. In the proposed design, the Octal Bus Transceiver SN74LVC4245A [25] is used for the necessary signal-level matching. Since certain (input) pins of the microcontroller are 5-V tolerant, not all signals need to be shifted.

To extend the measurement range, the chip can be connected to four external capacitors (one capacitor per channel and side). According to the manufacturer, the external integration capacitors should have low voltage coefficient, temperature coefficient, memory, and leakage current. The used 270 pF $\pm$ 2% Type AVX Corporation 06035A271GAT2A capacitors comply with these requirements and are therefore used in the design.

The integrator needs a system clock that drives all internal acquisition and conversion circuits. Depending on the chip's model, the nominal clock speed is either 10 or 15 MHz. In the design,

the 15 Mhz clock is generated by the microcontroller's internal PLL (Phased Locked Loop) circuit and level-shifted to 5 V.

A final important component for the correct function of the DDC is an external voltage reference, with a nominal voltage of $V_{Ref} = 4.096$ V. The integration capacitors are charged to $V_{Ref}$ at the beginning of each conversion cycle, and then depleted proportionally to the flowing current. It is very important that the reference is stable during the different operations, which are charging the capacitors and supplying reference charge to the internal ADC when converting. In the design, we decided to use a chip type REF6141 from Texas Instruments [26] because of its high current output, very fast load regulation, and good accuracy.

The chip offers a test mode that injects roughly 13 pC of (cumulative) charge into the selected range each time the IC's test-pin is toggled. The mode is used to check the system's health and calibration. It is performed every time the system boots up. If the check is successful, the system switches to normal measurement operation mode. Otherwise, an error is issued using a status LED.

While the chip performs very well and can easily be used, the DDC has some shortcomings which have to be addressed before using it for the targeted application; the range cannot be individually chosen per channel and the same applies to the integration time $T_{Int}$. Furthermore, the chip has a minimal integration time $T_{Int_{Min}}$ of 333 µs in continuous measurement mode, which results from the time needed to convert the data and additional overhead time for resetting the capacitors and side-switching. $T_{Int_{Min}}$ is dependent on the chips main clock and the 333 µs only apply for a system clock speed of 15 MHz. To address these shortcomings, we developed an optimized algorithm, which is presented below.

To provide the best flexibility and extensibility for future research, it was decided to implement a user-selectable sensor excitation voltage $V_{ex}$ for each sensor. It is generated by an external dual-channel 12-Bit digital-to-analog converter (DAC) type AD5627R [27]. Each of its two independent output channels is connected to an active second-order low-pass filter in Sallen-Key topology with a cutoff frequency of 5 Hz. The user can select voltages from 0.1 to 2.5 V with a 12-Bit amplitude resolution. The generated voltage is continuously read by the microcontroller's internal ADC for regulation and self-diagnostic purposes. The AD5627's internal high precision reference voltage is also used as an external voltage reference for the microcontroller's analog circuitry. The excitation voltage circuit is depicted on the left side of Figure 3.

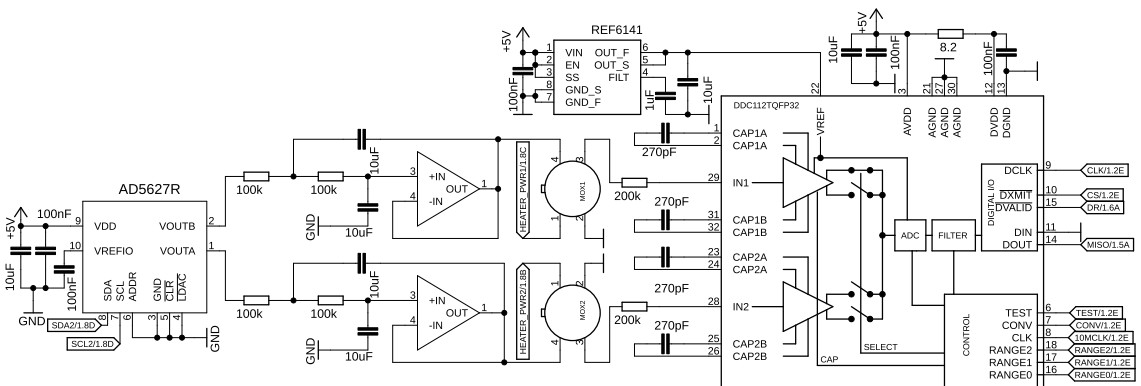

**Figure 3.** Schematic depicting (from left to right) the user-programmable sensor excitation voltage block (AD5627R with active low-pass filtering), two MOX sensors, the reference generator REF6141, and the DDC112 chip.

The central microcontroller of type STM32L433 is used to generate the integration signal using a PWM signal. It generates the main 15 MHz clock for the DDC112, sets the integration range (0–7) using three general purpose input–output (GPIO) lines, reads the digitized data from the DDC112 over SPI, converts the data to the corresponding resistance, and sends them to the

mainboard upon request. The controller's firmware also implements the aforementioned measurement algorithm, which manages best range selection, integration time optimization, and channel-switching. Furthermore, it supervises and regulates the two heater power supplies and the two sensor excitation voltages, generated by the AD5627R.

The firmware implements the readout algorithm; manages the heater control and supervision; stores system parameters, e.g., calibration data, sensor descriptions, etc.; and handles the internal system communication via I$^2$C. From the complex firmware functions, here, only the algorithms for the measurement and the calibration are described in the following.

Since the DDC chip has two independent channels, but only supports global integration time and range selection, having two possibly very different sensor values requires a dedicated algorithmic action to support such scenarios. The implemented algorithm is designed to always favor the highest possible integration time for low noise operation. This, however, directly affects the maximal sampling speed of the system if large resistances are connected. To achieve a minimal (continuous) sample rate of approximately 3 Hz, the highest integration time calculated by the algorithm is therefore limited to 100 ms.

Because of variances in absolute capacitance, variant thermal exposition, and slightly different trace lengths outside the IC package, the use of the external capacitors degrades the accuracy of the measured values and increases the noise. Hence, the algorithm tries to avoid the use of the external capacitors and only uses them if absolutely indispensable, which is the case for resistive values smaller than 100 Ω.

Since the DDC112 has a minimal integration time (in continuous mode) of $T_{IntMin} = 333$ μs and its maximal internal integration range $QFS_{max}$ is 350 pC, the minimal resistance that can be measured with a constant sensor excitation voltage $V_{ex} = 0.5$ V is given in Equation (1) and has to be greater than 476 kΩ. The minimal resistance that can be measured by using external capacitors $C_{ext}$ with ≈270 pC according to Equation (2) has to be greater than 157 kΩ. The fixed factor of 0.96 in the equation is a given by the hardware and allows the front end integrators to reach fullscale without having to completely swing to ground. To guarantee that the algorithm always terminates, a resistor satisfying Equations (1) and (2) $R_{inline} = 200$ kΩ is placed in series to the actual resistor that is to be measured.

$$
\begin{aligned}
R_{min} &\geq \frac{T_{IntMin} \times V_{ex}}{QFS_{max}} \\
R_{min} &\geq \frac{333 \text{ μs} \times 0.5 \text{ V}}{350 \text{ pC}} \\
R_{min} &\geq \approx 476 \text{ kΩ}
\end{aligned}
\tag{1}
$$

$$
\begin{aligned}
R_{min} &\geq \frac{T_{IntMin} \times V_{ex}}{0.96 \times C_{ext} \times V_{Ref}} \\
R_{min} &\geq \frac{333 \text{ μs} \times 0.5 \text{ V}}{0.96 \times 270 \text{ pF} \times 4.096 \text{ V}} \\
R_{min} &\geq \approx 157 \text{ kΩ}
\end{aligned}
\tag{2}
$$

$$
\begin{aligned}
R_{max} &\leq \frac{t_{IntMax} \times V_{ex}}{Q_{min}} \\
R_{max} &\leq \frac{100 \text{ ms} \times 0.5 \text{ V}}{10 \text{ pC}} \\
R_{max} &\leq 5 \text{ GΩ}
\end{aligned}
\tag{3}
$$

Although there is theoretically no upper limit for the maximal measurable resistance, the algorithms quality settings and physical board parameters, especially current leakage, in fact impose one. A minimum charge of 20% of the smallest integration range (10 pC) is required for a valid low noise measurement. Using the default excitation voltage of 0.5 V, the effective maximal resistance is given by Equation (3) and should be smaller than 5 GΩ. However, if required for research,

excitation voltage, as well as integration time, can be manually chosen by the user, therefore reaching much higher resistances.

The measurement algorithm is presented in the flowcharts in Figures 4 and 5. The system is initiated with the highest internal integration range (350 pC) and maximal integration time (100 ms). The continuous measurement is started by configuring the range (three GPIO lines representing a Bit pattern from 0–7), and then the first measurement is triggered by providing the integration signal (CONV) to the DDC112. It is generated by a 32-Bit timer of the microcontroller, operating in PWM (Pulse Width Modulation) mode. It produces a square wave with a fixed duty cycle of 50%, and its frequency is equal to half of the desired integration time. The PWM frequency change is encapsulated in a non-blocking subroutine called Start Measurement, which takes the desired integration time as a parameter.

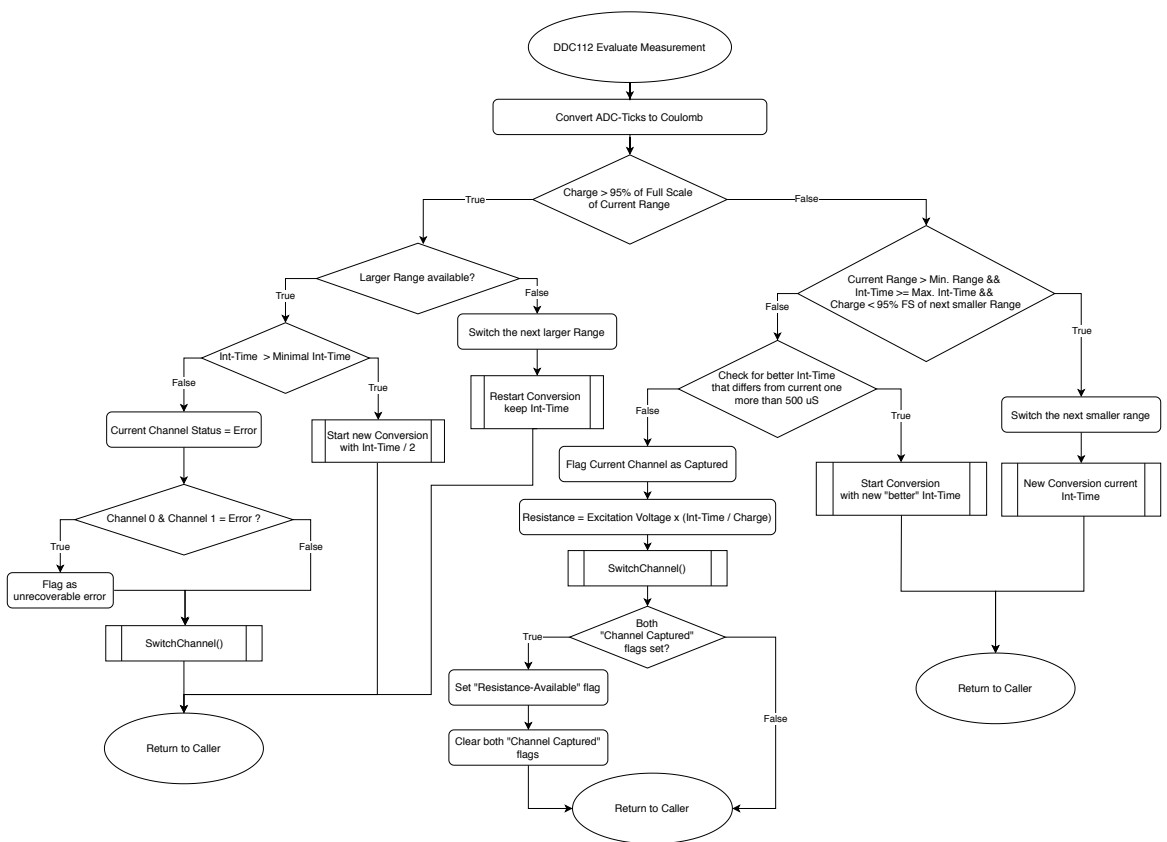

**Figure 4.** Simplified flow chart for the main measurement algorithm.

After the first measurement is started, the microcontroller enters the main loop where it waits for events such as data-ready event or master-read requests.

The data-ready event indicates that the DDC finished a conversion, which can now be downloaded. It is signaled to the microcontroller by the DDC112's DVALID line, generating an interrupt, which sets the data-ready flag. The event then triggers the data download over the SPI interface. In addition to the raw data, the integrator side (A or B) that converted the current value is stored (this information is obtained by the polarity of the PWM signal). Once the transmission completes, the DDC112 Evaluate Measurement function is called. The subroutine starts by converting the raw ADC ticks to a corresponding charge according to $Q = \frac{(ADC-Tick)-4096}{2^{20}} \times (1 + \frac{0.4}{100}) \times QFS$, where $QFS$ is the full scale (FS) charge ($Q$) in picocoulomb (pC) of the currently selected range. The acquired charge is then checked against the algorithms setting for optimal range at the highest possible integration time.

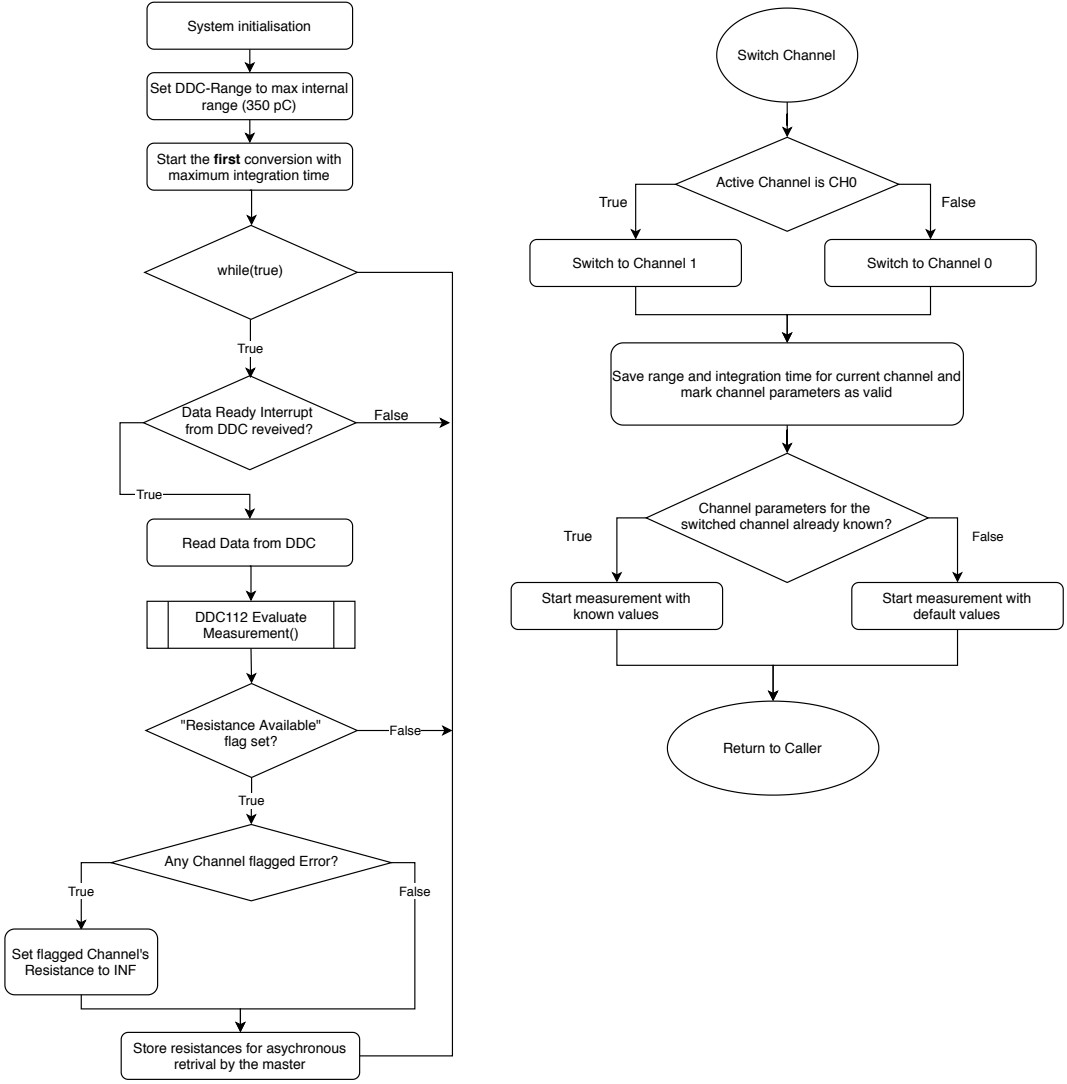

**Figure 5.** Simplified flow chart for the measurement algorithm helper functions.

If the acquired charge is larger than 95% *QFS* and a larger range is available, the range is changed and the measurement with the same integration time is repeated. The algorithm only starts reducing the integration time if no larger range is possible. Therefore, it checks if the current integration time is larger than the smallest integration time. If that condition is true, it reduces the time by half and repeats the measurement. If, however, the current measurement time is already equal to the smallest integration time, the channel is flagged as erroneous and the channel is switched. This should not happen as long as Equations (1) and (2) are satisfied.

If the charge is smaller than 95% of *QFS*, the algorithm checks if a smaller stage is available and that the charge is smaller than 95% *QFS* of that smaller range. Further, whether the current integration time is equal to the maximal time is checked. If all these conditions are met, the range is switched and the measurement is repeated. If only one of the conditions is not met, the else clause is executed, in which the integration time is reevaluated based on the acquired charge, and, if a better time that differs to the current one by more than 500 µs is found, a new measurement with that new integration time is initiated. Otherwise, the resistance is calculated, the channel marked as captured, and the non-blocking Switch Channel function called.

Following the return of the channel switching subroutine, the program checks if both channels have been captured. If this is the case, the global "Resistance Available" flag is set. If both channels have not been captured yet, the subroutine ends and returns to the main loop, where the status of the

"Resistance Available" flag is checked, and, depending on its state, the new values are prepared for asynchronous transmission to the master. Following that, a new measurement is always started and the cycle begins again.

In the case of the algorithm failing, the system is flagged with an unrecoverable error and informs the user. Such a scenario, for example, is triggered by not installing a sensor element so that there is no current to be integrated.

A calibration is performed every time the system starts up, or at user request. The calibration uses the DDCs Test pin (TP), which injects a fixed charge packet of $\approx$13 pC each time the voltage at the pin is toggled. It should be noted that there is a slight offset of about 0.2 pC between the two integrator Sides A and B, only occurring in the test mode. Because this is well known and documented by the manufacturer, it can be corrected by a simple subtraction. With our limited number of chips, we were able to measure a deviation of $\approx\pm$0.005 pC to the specified offset. Since the multiple injections are cumulative, they can be used to create multiple charge-packets for a multi-point calibration.

The charge injection is implemented by a timer in One-Pulse mode that is triggered by the timer that generates the integration signal CONV. Using the repetition counter option, the injection can be repeated as often as needed. The calibration is performed for each integration range at three points: low, medium, and high. Starting with the low point, always consisting of a single charge of 13 pC, followed by a several accumulating packages representing about mid-scale *QFS*, and finally a charge of about 90% of the range's full-scale, the calibration is performed. For each point the measured value is evaluated and correction factors, representing the deviation from the ideal injected values, are calculated and stored. These factors can later be used to correct the actual measurements. By default, the calibration and corrections are applied, this, however, can be changed by the user if desired.

In addition to the measurement, the firmware implements the I$^2$C communication with the master. The protocol is based on simple commands with optional data structures. Depending on the user request, either a full status packet with all available debug information or a small, and therefore fast, packet containing only the two resistive values along with their respective heater voltages and currents can be downloaded.

The firmware also implements a user page, a section in the CPU's flash memory that is kept, even if the firmware is updated. In this page, system information such as the module's I$^2$C address, parameters for the heater's PID regulation algorithm, sensor information (e.g., sensor name, installation date, etc.), and various other parameters are stored. Further, the quality settings for the algorithm, e.g., maximal integration time, setting to use the external integration range, etc. are stored here.

The external communication with the complete system is based on a Virtual COM Port (VCP) over the USB interface and is implemented in the firmware of the mainboard. The VCP option enables the system to be used without specific driver software on any operating system such as Windows, Linux, or macOS and the easy-to-use protocol provides a fast and seamless integration into existing measurement management tools such as LabView.

## 4. System Validation

To evaluate the system's readout performance, we tested the accuracy with 18 well known commercially available resistors with values from 10 $\Omega$ to 5 G$\Omega$. Each resistor was measured 500 times at room temperature ($\approx$22 °C) and $\approx$40% relative humidity. The minimum, maximum, mean, and standard deviation were calculated from these measurements. Ground truth measurements were take with a Keysight U1231A for resistances up to 60 M$\Omega$. For larger values, the manufacturer specified values were used. The resistors for the 10 $\Omega$ to 1 M$\Omega$ range are Yageo MFR series [28]. For the resistance values from 4 M$\Omega$ to 5 G$\Omega$, the components are Ohmite Slim-Mox [29]. All resistors have a specified accuracy of $\pm$1%.

The measurements were performed with the following system settings: maximal 100 ms of integration time, use of the external capacitors enabled, default algorithmic quality settings, and sensor

excitation voltage set to 0.5 V. From the resulting dataset, standard deviation, mean, and the coefficient of variation were calculated.

Figure 6 depicts the results from the accuracy measurement, as well as the coefficient of variation over the 9.6 decades. Figure 6b shows that the system is most error-prone in the lower two decades. The reason for this is the very short integration time and the use of the external capacitors (270 pF ± 2% Type AVX Corporation 06035A271GAT2A), which were used to measure the small resistors. As mentioned above, the external capacitors can have large deviations in their absolute capacitance, and their position outside the IC package makes them more sensitive to external influences such as temperature or electromagnetic interference. Since the system was initially designed to cover the range from 100 Ω upwards, these low ranges are not that important for the application and are only listed here for the means of presenting a more complete overview of the performance of the system. The system's average coefficient of variance over the complete range (0 Ω to 4 GΩ) is 5.41%. For the initially defined range (100 Ω to 4 GΩ), the system reaches an average accuracy of 0.27%.

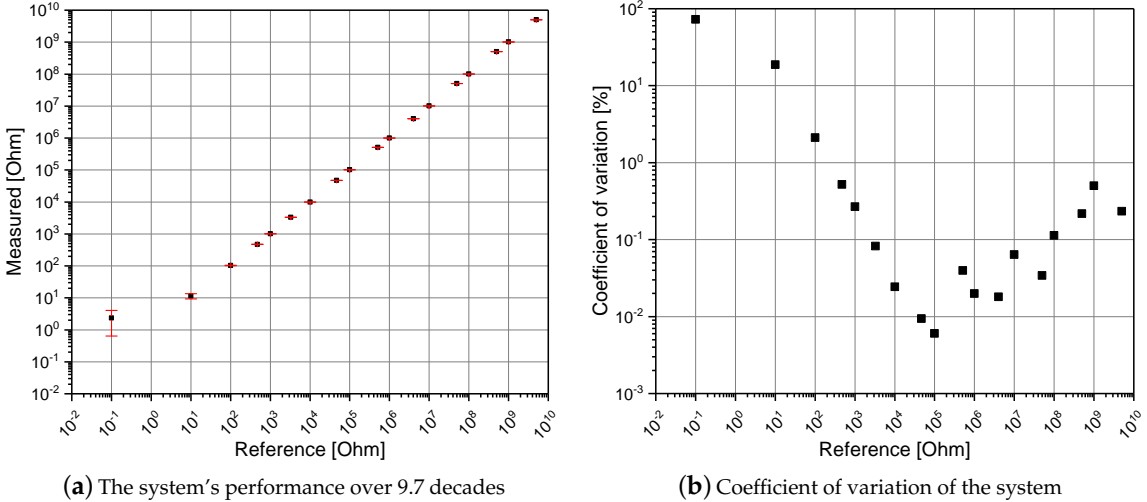

(**a**) The system's performance over 9.7 decades      (**b**) Coefficient of variation of the system

**Figure 6.** System performance over an input range of 9.7 decades (**a**) and the coefficient of variation for each decade (**b**).

## 5. Application Example

For the evaluation of the system's application usability, several tests in the form of an exemplary sensor evaluation (screening) were performed. These experiments are only used as proof of system concept and a more detailed analysis of the sensors will be published in an upcoming work.

In multiple test sets, the sensor's responses to volatile organic compounds (VOC) were investigated. The main goal was to determine whether the sensors, depending on their respective sintering temperature during the manufacturing process, could be used for reliable VOC detection. Additionally, the sensitivity to certain target gases (e.g., acetone) at different sensor temperatures was evaluated. The challenge that arises is the strongly varying resistance, due to the temperature of the semiconducting sensitive layer, as well as material changes that occur during the sensor production (e.g., grain growth). The described task directly maps to a typical optimization process [30–32].

The sensors were manufactured as follows: 10 µg tin oxide ($SnO_2$) were mixed with 100 µL water and treated in an ultrasonic bath for 1 h. The starting material was nanopowder [33] with a particle size of ≤100 nm. 1 µL of the suspension was dropped onto the sensor substrate (UST Umwelttsensortechnik GmbH type x33x), using a pipette, to create the sensitive layer.

A set of 16 sensors (combination of four sintering temperatures and four holding times) was created and examined afterwards. The sensor name is the combination of the sintering temperature and respective holding time. The set includes the sintering temperatures 700, 800, 900, and 1000 °C combined with the respective holding times of 10 min, 60 min, 12 h, or 24 h. After production,

all sensors were installed in a gas mixing unit (GMU) (see [17]) to which the presented system was connected.

The 16 custom sensors were then varied in the temperature range of 250 °C to 600 °C with an increment of 50 °C per step. At each temperature, the sensors were exposed to acetone test gas (5 ppm) for 25 min followed by a recovery time of 150 min, repeating these steps three times. After modifying the heater voltage, a settling time of 120 min was given to allow the sensor to reach its new temperature and to settle to a chemical equilibrium on the surface. The sensitivity S was calculated from the raw data as shown in Equation (4). $R_0$ and $R_x$ represent the mean value of the measured resistance over 2 min before and after analyte exposition at the end of each timestep.

$$S = \frac{R_0 - R_x}{R_0} \tag{4}$$

Figure 7a depicts the resistance variances of four selected sensors for all performed measurements. The sensors share the same sintering time but differ in sintering temperatures. Figure 7b presents the sensitivity S according to Equation (4), calculated from the absolute resistance values.

The result of this short study shows that the different manufacturing parameters of the $SnO_2$ substrate have enormous influences on the material properties. While the baseline resistances of the sensors do not show a dependence on sintering temperature and duration, the sensitivity of the sensors has a dependence on sintering temperature (Figure 7b). The operating temperature shows an optimum across all sensors in the range of 400–500 °C with sensitivities of up to 0.9, as depicted in Figure 8. A maximum in sensitivity is achieved for 700 °C and a holding time of 12 h. No significant results could be achieved for two sensors in the test (800 °C–1 h and 1000 °C–24 h) since their resistance could not be measured (>5 GΩ) reliably.

The successful measurement sets, however, showed a clear trend: excessive input of energy during the sintering process leads to a lowering of the sensor's sensitivity, which can be attributed to a coarsening of the polycrystalline structure of the semiconducting layer. The nanocrystalline raw material shows significant grain growth if the energy input is too high, which leads to a reduction of the active sensor surface. A similar effect is known and was described by Zhang et al. [31]

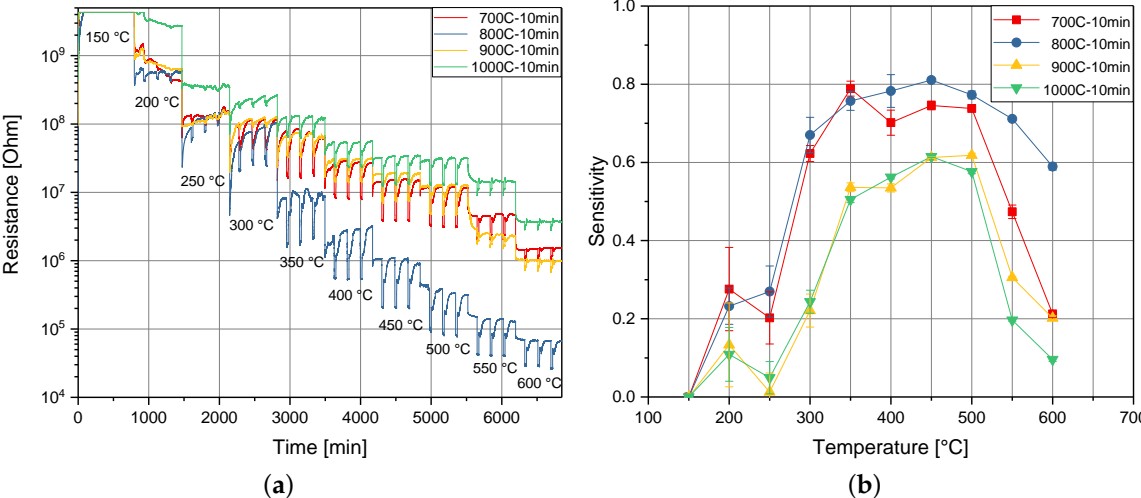

**Figure 7.** Restive changes of the sensors during the measurement program (**a**) and temperature dependency of the sensor sensitivity depending on the sintering temperature, all with a fixed sintering time of 10 min (**b**). The sensor name is the combination of the sintering temperature (°C) and respective holding time. The temperature information in (**a**) shows the working temperature of the sensors. All experiments were performed in synthetic air. Acetone was used as the test gas and diluted to 5 ppm with synthetic air while maintaining a total flow of 200 sccm.

The principle functionality of the water cooling was also evaluated in a separate experiment. The water cooling system was installed to reduce the heat in the aluminum of the measurement chamber, because, in an earlier version of the system, a noticeable influence between the temperature of the chamber and the measurement accuracy (especially when using the external integration capacitors) could be observed. The results for the tests of the water cooling are depicted in Figure 9. The columns of the plot represent the heating power summarized over all the 16 sensors. The column with the heating power of 9.6 W resembles the power for the sensors during the screening process described in the work. Using the water cooling, the temperatures on the surface as well as on the inside of the chamber could be reduced to such a degree that the chamber temperature no longer influences the measurements.

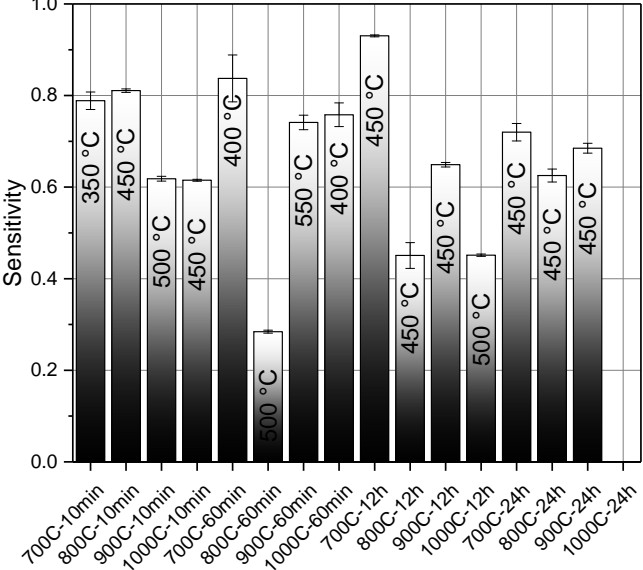

**Figure 8.** Maximum sensitivity S with error. Within the bars, the temperature at which the maximum was reached is given.

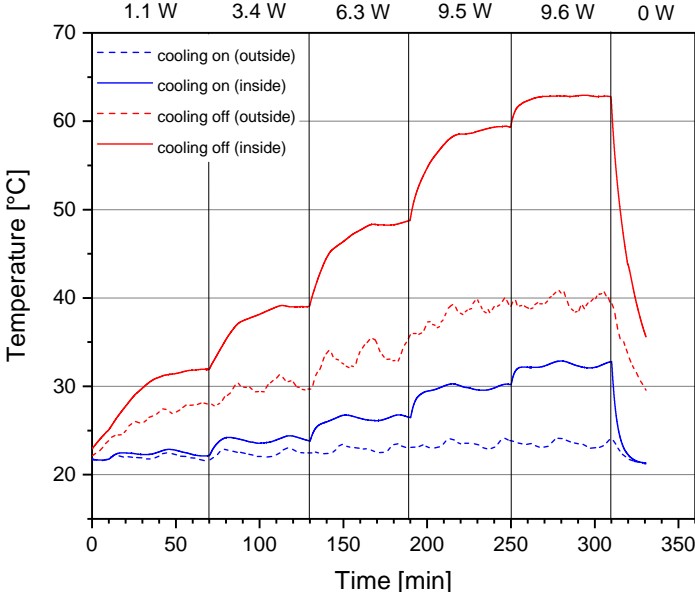

**Figure 9.** Effects on the temperatures inside and outside of the measurement chamber, with and without water cooling enabled, at different levels of sensor-heating.

## 6. Conclusions

In this paper, we present a compact, modular embedded system, built with commercially available electronic parts that provides an easy to manage, flexible, and extensible base for the use of up to 16 different MOX gas sensors. A sensor module provides two independent complete interfaces for MOX gas sensors, each consisting of a low ripple, high efficiency heater power supply and a high range resistive readout circuit.

The presented system can be used in multiple application scenarios, e.g., screening of newly developed sensor substrates or for specific target gas detection as a multi-sensor array. The open communication protocol and versatile USB interface enables fast and easy integration into existing or new setups. The results in Section 5 clearly show that even a simple screening experiment requires a very large dynamic measurement range from several k$\Omega$ up to 4 G$\Omega$.

With a wide resistive input range of 9.7 decades and an average system error of 0.27% in the range from 100 $\Omega$ to 5 G$\Omega$, the system allows a large spectrum of MOX sensor types to be used in any combination. The implemented algorithm for measurement quality can be user-customized for more speed or even higher resistances than 4 G$\Omega$, and, along with the digitally programmable sensor excitation voltage, the system provides the best possible flexibility and extensibility for use as a reliable base for future research in the field of MOX gas sensors.

**Author Contributions:** C.H. is the lead author and was responsible for the system design, hardware construction, software development, resistance measurement experiments, and writing the final paper. J.W. is co-author of this work. He designed, executed, and evaluated the application example presented in Section 5 and provided help for the literature research. S.M. was tasked with design and manufacturing of the measurement chamber, required for the final system. P.K. and N.J. are the directors of the Institute of Safety and Security Research (ISF). R.T. and N.J. are the referees in the proceedings of C.H. They all contributed to the work by providing design advice, experimental setup guidance, project coordination, and several iterations of paper review. All authors have read and agreed to the published version of the manuscript.

**Funding:** This research received internal funding from the Institute of Safety and Security Research at BRS-U.

**Conflicts of Interest:** The authors declare no conflicts of interest.

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
