# Peer review of "A Compact 16 Channel Embedded System with High Dynamic Range Readout and Heater Management for Semiconducting Metal Oxide Gas Sensors"

_electronics, doi:10.3390/electronics9111855_

Round 1
Reviewer 1 Report
Line 175 "the" (beginning of the sentence) should be deleted.
Line 194 "rage" is probably range.
In (1) tmin is actually TIntMin and QFS is defined only at next page.
In (2) Cext is not defined.
All variables and parameters should be defined.
Figure 4 is too small and has to be enlarged by splitting to 2 or 3 figures.
Line 337 "sensitivities of up to 0.9" probably of sensors not presented in the graph.
Section 5 Application Example could be extended by presenting also 2 additional graphs for the 4 sensors 700-(10min-24h) and by a different concentration of the acetone test gas.
Line 365 "That a even", delete a. Also chapter 5 probably means section 5.
Reviewer 2 Report
The submi9tted manuscript is an interesting review of the designed and built gas sensing set-up. Its publication should be interesting for the readers but requires some necessary amendments.
Here are my detailed remarks:
- You should give more information about the consumed energy. Moreover, it would be interesting to monitor temperature inside the gas chamber to be sure that the applied water cooling is sufficient. Please add some information about it.
- p. 6 you claim that there is no upper limit of the measured resistance. So high DC resistances as GigaOhms require special PCB boards. Otherwise you will have some leakage currents similar to the currents used for measurements. Please give some comments about it.
- Fig. 4 can't be read. You have to show it in a different way or reduce the number of diagrams.
- p. 9 you have measured some prototype gas sensors. Please add more description about the sensors (their thickness, working area etc.)
- You have used 5 ppm of acetone. Was it dilluted in synthetic air?
- Fig. 6 you have to add in the caption information about ambient atmosphere.
- There is a company jlm innovation: https://www.jlm-innovation.de/about producing the systems for an array of commercial gas sensors. Please comment their solutions and compare to your.
- You have mentioned that inherent noise is a problem in your system. Please give some data about noise intensity. You should comment another electronic circuit solution presented in: Kotarski, M., & Smulko, J. (2009). Noise measurement set-ups for fluctuations-enhanced gas sensing. Metrol. Meas. Syst, 16(3), 457-464. where flicker noise was measured to gather more information about ambient gas.
Reviewer 3 Report
The automatization of measurement process (for gas sensors with different parameters) may improve the development on new types of sensors, as well as monitoring of manufacturing parameters. From this point of view the manuscript might be interesting for wide group of people working with metal oxide sensors.
The authors approached the development of the sensor module responsibly. The block diagram of the measurement algorithm is described quite clearly. The system was checked thoroughly at room temperature. However, it is not clearly defined distances between sensors (in one module and neighbor modules). It would be interesting to know if there is any influence of the heating of the adjacent sensor.
From the formal point of view the manuscript is well organized and presented in clear form. The parts describing hardware and software are extensive and detailed. On the other hand, the chapter 5 where were reported testing of exemplary sensors leaves some questions open. There is describing that 16 sensors were measured. However, it is not clear if sensors were used singly or in pairs. It is good to know if measured results are independent of sensor position (MOS1 MOS2 in fig 2). Undoubtedly, the presented data are sufficient to confirm that designed sensor module is fully functional. But, to my opinion more results on exemplary sensor would make the publication more valuable.
Also, there is a mistype in fig. 2 right: “heating MOX” 1 is twice.
